# Renewable energy as a solution to climate change: Insights from a comprehensive study across nations

Keshani Attanayake[1], Isuru Wickramage[1], Udul Samarasinghe[1], Yasangi Ranmini[1], Sandali Ehalapitiya[1], Ruwan Jayathilaka[2]*, Shanta Yapa[3]

1 SLIIT Business School, Sri Lanka Institute of Information Technology, Malabe, Sri Lanka, 2 Department of Information Management, SLIIT Business School, Sri Lanka Institute of Information Management, Malabe, Sri Lanka, 3 SLIIT Business School, Sri Lanka Institute of Information Management, Malabe, Sri Lanka

* ruwan.j@sliit.lk

## Abstract

Without fundamentally altering how humans generate and utilise energy, there is no effective strategy to safeguard the environment. The motivation behind this study was to analyse the effectiveness of renewable energy in addressing climate change, as it is one of the most pressing global issues. This study involved the analysis of panel data covering 138 nations over a 27 year period, from 1995 to 2021, making it the latest addition to the existing literature. We examined the extent of the impact of renewable energy on carbon dioxide over time using panel, linear, and non-linear regression approaches. The results of our analysis, revealed that the majority of countries with the exception of Canada, exhibited a downward trend, underscoring the potential of increasing renewable energy consumption as an effective method to reduce carbon dioxide emissions and combat climate change. Furthermore, to reduce emissions and combat climate change, it is advisable for nations with the highest carbon dioxide emissions to adopt and successfully transition to renewable energy sources.

## Introduction

The rising challenges of energy production and climate change necessitate a transition towards Renewable Energy Sources (RES) to mitigate carbon emissions and ensure a sustainable future [1–3]. According to the Population Reference Bureau, the world population is predicted to expand from 7.8 billion in 2020 to 9.9 billion by 2050, which requires a staggering 80% more energy [4]. This will increase the worldwide demand for energy and related energy services to support fundamental human requirements and serve industrial activities to satisfy social and economic development [5]. Since the early 1800s and the industrial revolution, the global usage of fossil fuels (coal, oil, and gas) to satisfy the increasing energy demand has resulted in a significant increase in Carbon Dioxide ($CO_2$) emissions which is the fundamental driver of climate change and global warming [6]. Thus, the adverse environmental impacts, as well as the factors which influence $CO_2$ emissions have remained an important subject of worldwide discussions among the academic community for many years [7, 8]. These concerns ignited the

**Data Availability Statement:** The data underlying the results presented in the study are available from the World Bank Open Data datasets at the following two links: https://data.worldbank.org/indicator/EG.FEC.RNEW.ZS and https://data.worldbank.org/indicator/ST.INT.ARVL.

**Funding:** The authors received no specific funding for this work.

**Competing interests:** The authors have declared that no competing interests exist.

motivation to conduct this study to identify the effectiveness of Renewable Energy (RE) usage on reducing the dependency on fossil fuels which allows a significant decrease in $CO_2$ emissions and therefore mitigating the impacts it has on the environment.

The over-dependence on fossil fuels to fulfil the energy demand has led to energy shortages and a rapid increase in $CO_2$ emissions, which causes significant adverse effects on global climate as fossil fuels account for more than 75% of all Green House Gas (GHG) emissions and almost 90% of all $CO_2$ emissions, which is identified as the most significant cause of climate change in the world [9]. By 2040, worldwide energy consumption is predicted to increase by 56%. If the usage of fossil continues resulting further increase of $CO_2$ emissions [10] it can cause substantial harm to the environment and the living organisms. Therefore, switching from traditional energy to RE is of utmost importance. The International Kyoto Protocol has requested a 60% reduction in carbon emissions by 2050 [11]. Consequently, the academic community has conducted numerous studies on achieving sustainability goals, carbon neutrality, and exploring alternative options for fossil fuels, such as natural gas and RE, along with their impact on environmental sustainability [12–16].

Enforcing strict environmental regulations, producing energy using Renewable Energy Technologies (RET) and optimizing Renewable Energy Consumption (REC) using effective and sustainable approaches should be key focus areas of governments [17]. Similarly, they must stop relying on fossil fuels and invest in reliable, clean, accessible, and affordable alternative energy sources. Hence, minimising the usage of fossil fuel and optimising the use of RE would be the optimal option to attain carbon neutrality and sustainably fulfil the energy needs of humans as it can reduce $CO_2$ emissions [18–20].

Transitioning to RE for energy production is essential as it helps reduce carbon emissions, mitigate climate change, enhance environmental sustainability, and foster long-term socio-economic benefits such as sustainable economic growth [21–23].

Derived from solar, geophysical, or biological resources, RE is regenerated at a rate equal to or greater than its rate of consumption using natural processes such as biomass, solar energy, wind energy, geothermal heat, hydropower, tide and waves, and ocean thermal energy [24]. A significant and negative correlation between Renewable Energy Production (REP) and carbon emissions has been identified by multiple scholars [25, 26]. With the advancement of technology, REC is becoming more cost-effective and environmentally sustainable. Reducing prices have made RE more suitable for countries with limited resources in which we will be able to see a significant increase in demand for RE in the future.

Further, there is a strong possibility that a large portion of the new power supply during the upcoming years will come from RES due to reduced costs [27]. However, transitioning to REP presents significant challenges for developing and least developed countries. Higher capital investments, technological requirements, and government intervention in developing new policies and strategies, as well as facilitating research and development, are significant in transitioning to REP [28]. Governments and other relevant stakeholders initiate RE related projects to smoothen the transition from non-RE to RE in certain industries. It has been proven that stakeholder engagement and their relationship with project management teams contributes significantly to the success of the project in countries such as Pakistan [29]. A high success rate for such projects will directly impact the $CO_2$ emission levels around those industries.

This study contributes to the existing literature in several fronts. Firstly, it analyses each country separately and categorises them based on their level of economic development as developed, developing, least developed, and transitional economies. Despite extensive research on the relationship between $CO_2$ emissions and RE of developed and developing economies, a significant gap in the literature exists regarding the context of least developed and transitional economies. Focusing on understudied areas, our research aims to contribute to this limited

body of literature by addressing the context of least developed and transitional economies to provide valuable insights for policymakers. Secondly, panel regression, linear regression and non-linear regression techniques are used in this study, allowing for a more comprehensive analysis of both linear and non-linear relationships between $CO_2$ emissions and RE and the level of the impact on the respective variables over a period. This is important in understanding how the relationship between $CO_2$ emissions and RE has evolved across different countries and accurately identifying patterns and changes. Thirdly, this study visualises the top 12 $CO_2$ emitting countries, and the relationship between REC and $CO_2$ emissions in each country in a single graph allowing the users to identify the countries who negatively impacts the environment the most and how they can reduce those effects by transitioning to RES. Finally, this study represents the latest addition to the current literature, analysing panel data from 1995 to 2021 over 27 years covering 138 nations. While previous research has focused on individual countries or a limited number of countries, there need to be more comprehensive global studies to provide a thorough understanding of the worldwide relationship between $CO_2$ emissions and RE for a significant period. Therefore, this study, satisfying the research question to understand what the impact of RE on $CO_2$ emissions is, provides a broader analysis based on a more substantial number of countries considering a more comprehensive period using the latest available data to obtain a more accurate depiction of the global status of the relationship between two variables helping for more precise decision-making.

The study's primary objective is to investigate the impact of RE on $CO_2$ emissions in various countries to make the said four contributions to the existing literature while identifying the countries who negatively impact the environment due to their high $CO_2$ emissions and how they can reduce such impacts by transitioning towards RES. As the interrelationship and overall social impact between $CO_2$ emissions and RE remain to be fully understood by the industry professionals, policymakers, and academic community despite the substantial amount of research, this study will provide accurate and reliable insights to make informed decisions regarding the promotion of RE and mitigate $CO_2$ emissions tailored to each country's unique context.

The structure of this paper is as follows: in the initial section, a comprehensive review of the existing literature is provided. The following section introduces the methods used for data collection. Next, the results are presented, followed by a detailed discussion section. The paper's final section summarises the study's findings and policy recommendations.

## Critical review of existing literature

The method followed to identify relevant research articles from past literature is illustrated in (Fig 1). To ensure high quality and relevance to the study's objectives, reputable academic journals such as ScienceDirect, Emerald Insight, SAGE journals Online, Taylor & Francis Online, Springer Link, and Wiley Online Library were used to identify the most suitable articles and yielded 318 publications. Through careful screening, 160 articles were excluded due to insufficient relevance or mismatched information. From the remainder, 158 articles were selected based on the title, keywords, and abstract. Finally, 98 publications that align with the study's focused area were chosen based on the H-index and quartile rating of the published journals. The selected articles were again categorised as developed, developing, transitional, and least developed economies.

The subsequent section emphasises the contributions of scholars to the existing literature on the correlation between $CO_2$ emissions and RE across different countries and economies, laying a robust scientific groundwork for this study. The results are classified based on the four categories of economic development.

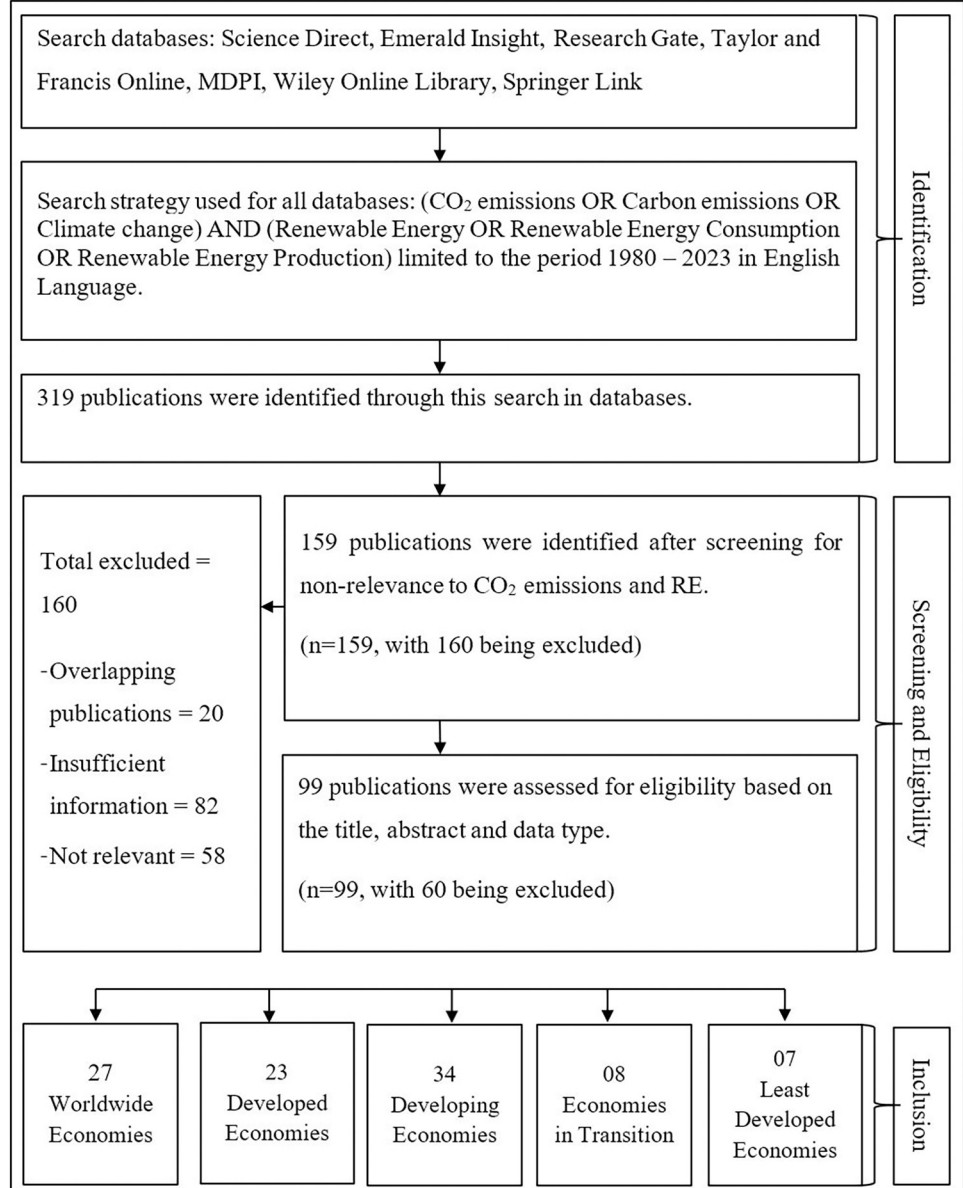

**Fig 1. Literature search flow diagram.** Source: Authors' illustrations.

## World economies

Having explored the relationship between $CO_2$ emissions and RE among multiple countries in Asia, Africa, Europe, Oceania, and North and South America, scholars have concluded that implementation of RE among all nations helps to reduce reliance on fossil fuels and mitigate climate change [30, 31]. Similar studies based on multiple European countries also identified that REC helps minimise $CO_2$ emissions [32–34]. Multiple studies based in African countries also identified RE as an efficient replacement for fossil fuels which helps to reduce carbon emissions and mitigate environmental problems [35, 36]. Another study focusing on multiple countries in the Middle East and North Africa (MENA) identified a unidirectional causality from RE to $CO_2$ emissions in the short run and unidirectional causality from $CO_2$ emissions

to RE in the long run [37]. Moreover, multiple studies based on Organization for Economic Cooperation and Development (OECD) countries concluded that REC and investment in RE reduces $CO_2$ emissions while helping to achieve environmental sustainability [16, 38, 39]. Additionally, multiple studies on G7 countries and Brazil, Russia, India, China, and South Africa (BRICS) countries have identified that RE helps significantly mitigate $CO_2$ emissions [40, 41]. Moreover, a study in Europe and Central Asia published that when REP increases by 10%, $CO_2$ per capita emissions decrease by 4.1% [25]. However, the preceding research findings cannot be generalised to all countries and regions worldwide. These studies focused on a few countries in Asia, Africa, America, Europe, and the Middle East without investigating the combined effect.

## Developed economies

Various scholars have explored the relationship between $CO_2$ emissions and RE among developed economies. Studies based on developed economies namely Canada, France, Japan, Netherlands, Spain, Sweden, Switzerland, United Kingdom, and the United States have concluded that RE reduces $CO_2$ emissions and the use of RE is beneficial for environmental protection [42, 43]. Further, multiple studies about $CO_2$ emissions and RE in the United States collectively concluded that despite the rising public acceptance of the fact that burning of fossil fuels is a major reason for climate change and the usage of RE can mitigate climate change, the levels of REC of the United States cannot significantly contribute to $CO_2$ emissions reduction yet [44–49]. Similarly, studies among countries in Europe identified that $CO_2$ emissions have a positive impact on climate change in the long run, and RE can be used to minimise the effects of climate change [50–52]. A few studies based in Belgium, France and Germany identified that RE usage reduces $CO_2$ emissions, GHG emissions and reduces energy costs in developed economies [53, 54]. Moreover, studies which examined the relationship between $CO_2$ emissions and RE in Austria, Japan, and New Zealand concluded that REC helps to reduce $CO_2$ emissions, mitigate climate changes and protect the environment [55, 56]. Multiple studies based on UK, Spain and Switzerland expressed that these countries have invested in RE to reduce $CO_2$ emissions and balance the energy demand [57, 58]. However, the above studies were conducted for the selected few countries and the majority were not conducted using the latest available data making them less applicable to the current situation.

## Developing economies

Findings of the empirical studies based on developing countries reveal that the growth of $CO_2$ emissions is substantial and rapidly increasing among developing economies [59]. Despite being recognised as the world's largest $CO_2$ emitter, China must focus on reducing its $CO_2$ emissions by utilising RE. Multiple studies have concluded that the adoption of RE has a significant negative impact on $CO_2$ emissions in China, contributing to the mitigating climate change [60–63]. Similarly, India has ranked as the world's fourth-largest $CO_2$ emitter, followed by the United States and the European Union [64]. Multiple studies have identified the implementation of RES as a sustainable power supply to mitigate the negative environmental impact due to fossil fuel and reduce $CO_2$ emissions and climate change in India [65]. Studies in India, Pakistan, Nepal, and Sri Lanka have revealed that RE improves air quality and mitigates $CO_2$ emissions in these countries, although the share of electricity produced using RES is comparatively low compared to other developing economies [66–68]. Similar studies in China, Pakistan, Malaysia, and Indonesia have revealed that increasing and actively funding the proportion of power generated from RES is crucial to support sustainable development, climate change mitigation, and environmental protection [37, 63, 69, 70].

Further, studies analysing the relationship between $CO_2$ emissions and RE among multiple Middle Eastern countries such as Jordan, Iran and Turkey, concluded that governments must further invest in the implementation of RET as it helps to lower the $CO_2$ [71–73]. Similarly, few studies proved that RE helps to mitigate climate change and a negative relationship between the two variables was identified in multiple countries among Northern and Southern American countries such as Argentina, Brazil and Mexico [74–76]. Further, studies based on African regions have concluded that rising $CO_2$ emissions contribute significantly to climate change and encourage REC as it negatively affects $CO_2$ emissions [77–80]. Although the studies mentioned above were published recently, most were not based on the latest available data and were limited to single or multiple countries in a region. This would not give an accurate result regarding the current $CO_2$ emissions and REC status among developing economies.

## Economies in transition (transitional economies)

There needs to be more studies analysing the relationship between $CO_2$ emissions and RE among transitional economies. A study on multiple transitional economies identified a negative relationship between $CO_2$ emissions and RE and concluded that REC must be increased to reduce $CO_2$ emissions among transitional economies [81]. Similar studies on Russia, Azerbaijan, Kyrgyzstan and multiple transitional economies concluded that the burning of fossil fuels causes an increase in $CO_2$ emissions, while $CO_2$ emissions have a negative relationship with REC [82–84]. Furthermore, multiple studies have discussed the importance of increasing the share of RE generation among transitional economies in terms of energy security, economic and social contexts [85–88]. However, the above studies were conducted for a few transitional economies, and the lack of studies based on the latest available data highlights the importance of further analysing the relationship between $CO_2$ emissions and RE among transitional economies.

## Least developed

Similar to transitional economies, there needs to be more research on the link between $CO_2$ emissions and RE in least developed economies. A unidirectional causality from $CO_2$ emissions to RE has been identified in Bangladesh, and it has been concluded that RE can be used to minimise environmental pollution [89]. A similar study based in Nepal identified that a 01% increase in REC reduces $CO_2$ emissions by 0.41% in the long run. Nepal has significantly lowered GHG and $CO_2$ emissions by implementing RET [90]. Moreover, multiple studies have shown that the implementation of RET is helpful to least-developed countries to mitigate climate change, decrease pollution, serve increasing energy demand and electrification in rural areas [91–95]. However, the limited number of studies highlights the gap in extant literature, which analyses the correlation between $CO_2$ emissions and RE among least-developed economies using the latest data.

In summary, despite the existing body of literature investigating the relationship between $CO_2$ emissions and REC, studies covering the global situation, multiple countries or regions with the economic development category are insignificant. Furthermore, there exists a need for more research on transitional and least developed economies.

## Data and methodology

The following section presents a detailed description of the variables employed in this study, the data sources through which data was collected and the statistical methods used to compute the results.

## Data

This section provides a detailed view of the data gathered from secondary data sources. REC as a percentage of Total Energy Consumption (% of TEC) from the World Bank Open Data was employed from 1995 to 2021. To measure the amount of carbon emissions, the data on Annual $CO_2$ emissions from 1995 to 2021 was used from the data sources published by Our World in Data. Through the data collection process, 138 countries were selected subject to the data availability. They were categorised according to the economic development category as developing, developed, least developed and economies in transition. In 27.5% of the cases, the countries were developed, 44.9% were developing, 8.7% were transitional economies, and 18.8% were least developed. Forecasts were made to fill in any missing data by utilising simple linear regressions. Both backward and forward forecasts were used for this purpose. S1 Appendix presents the data file used for this study.

## Methodology

This section reveals the methodologies that were employed to derive the results. A summary statistics table was utilised to display the preliminary results of the data used in this study.

F test, Hausman test and Breusch Pagan LM tests were conducted to decide which specification test to be used out of the Fixed Effect Model (FEM), Random Effect Model (REM) and Pooled Ordinary Least Square (POLS) Model for four economic development categories and the worldwide category separately. The equation below was used to conduct the panel regression test for five categories separately.

$$Y_{i,t} = \beta_0 + \beta_1 X_{i,t} + \varepsilon_{i,t} \tag{1}$$

Thereafter, linear and non-linear regressions were used to analyse the impact of renewable energy on climate change for individual countries using the equation below:

$$Y_{i,t} = \beta_0 + \beta_1 X_{i,t} + \beta_2 X^2_{i,t} + \beta_3 X^3_{i,t} + \beta_4 X^4_{3i,t} + \varepsilon_{i,t} \tag{2}$$

where $Y$ denotes $CO_2$ emissions, $X$ denotes REC, $I$ denotes the country ($i = 1,\ldots N$), $t$ denotes the time period ($t = 1,\ldots T$) and $\varepsilon_{i,t}$ refers to the error term.

## Empirical results and discussion

The descriptive statistics for the two variables, REC as a percentage of TEC and $CO_2$ emissions are shown in S2 Appendix. S3 and S4 Appendices shows the difference between the averages of $CO_2$ emissions and REC as a percentage of TEC from 1995–2004 period and 2012–2021 period. Table 1 highlights the highest 40 $CO_2$ emitting countries for the period 1995–2004 and 2012–2021, including the percentage change while Table 2 depicts the lowest 40 countries consuming RE for the same periods providing in-depth insights of how countries across the globe uses RES and the level of $CO_2$ emissions they emit.

According to Table 1, the US has the highest $CO_2$ emissions among the developing countries in both 1995–2004 and 2012–2021 periods. However, from 1995–2004 to 2012–2021, we can see a 10% reduction which has resulted in 5255.338 million of tons. Even though developed economies generally tend to produce more $CO_2$, an interesting fact is that China and India can be identified as the countries with the highest and the 3rd highest average $CO_2$ emissions in the world in the 2012–2021 decade. This is due to large scale manufacturing initiatives they are carrying out [96]. Interestingly, Russia is also among the top 5 $CO_2$ emitting countries despite being a transitional economy.

**Table 1. Countries with the 10 highest averages for $CO_2$ emissions in each economic development category.**

| Country | Average of $CO_2$ from 1995–2004 | Average of $CO_2$ from 2012–2021 | Percentage Change | |
|---|---|---|---|---|
| **Developed Countries** | | | | |
| United States | 5829.835 | 5255.338 | -10% | ▼ |
| Japan | 1254.93 | 1186.107 | -5% | ▼ |
| Germany | 914.855 | 759.437 | -17% | ▼ |
| Canada | 544.487 | 566.661 | 4% | ▲ |
| Australia | 343.515 | 404.915 | 18% | ▲ |
| United Kingdom | 569.814 | 403.067 | -29% | ▼ |
| Italy | 468.856 | 351.406 | -25% | ▼ |
| France | 405.993 | 327.103 | -19% | ▼ |
| Poland | 335.915 | 322.291 | -4% | ▼ |
| Spain | 301.802 | 256.525 | -15% | ▼ |
| **Developing Countries** | | | | |
| China | 3881.647 | 10290.07 | 165% | ▲ |
| India | 944.875 | 2365.841 | 150% | ▲ |
| Iran | 351.723 | 665.255 | 89% | ▲ |
| Saudi Arabia | 280.948 | 630.582 | 124% | ▲ |
| Indonesia | 277.975 | 564.412 | 103% | ▲ |
| Brazil | 324.976 | 499.153 | 54% | ▲ |
| Mexico | 392.025 | 465.248 | 19% | ▲ |
| South Africa | 382.079 | 450.594 | 18% | ▲ |
| Turkey | 215.889 | 397.067 | 84% | ▲ |
| Thailand | 182.775 | 285.244 | 56% | ▲ |
| **Economies in Transition** | | | | |
| Russia | 1521.989 | 1664.016 | 9% | ▲ |
| Kazakhstan | 158.141 | 290.438 | 84% | ▲ |
| Ukraine | 321.009 | 240.219 | -25% | ▼ |
| Uzbekistan | 117.724 | 113.643 | -3% | ▼ |
| Belarus | 57.836 | 60.884 | 5% | ▲ |
| Azerbaijan | 30.523 | 35.302 | 16% | ▲ |
| Georgia | 3.962 | 9.788 | 147% | ▲ |
| Kyrgyzstan | 5.087 | 9.697 | 91% | ▲ |
| North Macedonia | 11.428 | 7.399 | -35% | ▼ |
| Tajikistan | 2.347 | 6.875 | 193% | ▲ |
| **Least Developed Economies** | | | | |
| Bangladesh | 27.2878 | 77.6750 | 185% | ▲ |
| Myanmar | 8.9274 | 24.8839 | 179% | ▲ |
| Angola | 10.4593 | 23.8687 | 128% | ▲ |
| Sudan | 5.8846 | 19.8306 | 237% | ▲ |
| Ethiopia | 3.6709 | 14.1438 | 285% | ▲ |
| Cambodia | 1.9503 | 11.9984 | 515% | ▲ |
| Tanzania | 2.9742 | 11.1960 | 276% | ▲ |
| Senegal | 3.9976 | 10.4994 | 163% | ▲ |
| Nepal | 2.6979 | 10.3971 | 285% | ▲ |
| Mozambique | 1.3783 | 6.4031 | 365% | ▲ |

Source: Authors' calculations based on data from World Bank and Our World in Data.

**Table 2. Countries with the 10 lowest averages for renewable energy consumption in each economic development category.**

| Country | Average of REC from 1995–2004 | Average of REC from 2012–2021 | Percentage Change | |
|---|---|---|---|---|
| **Developed Countries** | | | | |
| Malta | 0.031 | 6.173 | 19813% | ▲ |
| Japan | 3.907 | 6.307 | 61% | ▲ |
| Netherlands | 1.684 | 6.366 | 278% | ▲ |
| United Kingdom | 0.986 | 8.816 | 794% | ▲ |
| Macao | 0.801 | 9.089 | 1035% | ▲ |
| Australia | 8.264 | 9.316 | 13% | ▲ |
| Ireland | 2.067 | 9.511 | 360% | ▲ |
| Belgium | 1.489 | 9.599 | 545% | ▲ |
| United States | 5 | 9.713 | 94% | ▲ |
| Cyprus | 3.229 | 10.794 | 234% | ▲ |
| **Developing Countries** | | | | |
| Saudi Arabia | 0.012 | 0.015 | 25% | ▲ |
| Algeria | 0.461 | 0.115 | -75% | ▼ |
| Hong Kong | 0.405 | 0.176 | -57% | ▼ |
| Singapore | 0.482 | 0.662 | 37% | ▲ |
| Iraq | 0.358 | 0.95 | 165% | ▲ |
| Iran | 0.758 | 0.981 | 29% | ▲ |
| Maldives | 2.47 | 1.109 | -55% | ▼ |
| Seychelles | 1.448 | 1.205 | -17% | ▼ |
| Bahamas | 1.324 | 1.277 | -4% | ▼ |
| Syria | 1.868 | 1.485 | -21% | ▼ |
| **Economies in Transition** | | | | |
| Uzbekistan | 1.192 | 1.496 | 26% | ▲ |
| Kazakhstan | 2.006 | 1.64 | -18% | ▼ |
| Azerbaijan | 2.19 | 2.26 | 3% | ▲ |
| Russia | 3.613 | 3.27 | -9% | ▼ |
| Ukraine | 1.165 | 5.225 | 348% | ▲ |
| Belarus | 4.759 | 7.385 | 55% | ▲ |
| North Macedonia | 16.533 | 20.43 | 24% | ▲ |
| Kyrgyzstan | 31.052 | 23.893 | -23% | ▼ |
| Moldova | 4.722 | 24.913 | 428% | ▲ |
| Georgia | 49.771 | 27.87 | -44% | ▼ |
| **Least Developed Economies** | | | | |
| Mauritius | 26.1822 | 8.8932 | -66% | ▼ |
| Eritrea | 36.9926 | 19.1706 | -48% | ▼ |
| Bangladesh | 58.0548 | 29.9455 | -48% | ▼ |
| Senegal | 47.0836 | 39.7582 | -16% | ▼ |
| Lesotho | 54.0377 | 41.8866 | -22% | ▼ |
| Benin | 73.9628 | 45.6782 | -38% | ▼ |
| Angola | 71.4310 | 49.9982 | -30% | ▼ |
| Gambia | 59.6694 | 51.3397 | -14% | ▼ |
| Cambodia | 81.4842 | 58.5645 | -28% | ▼ |
| Sudan | 79.7331 | 61.2434 | -23% | ▼ |

Source: Authors' calculations based on data from World Bank and Our World in Data.

On the other hand, the least developed countries have significantly lower amounts of $CO_2$ emissions, whereas the highest $CO_2$ emitting country, Bangladesh, emits only 77.675 million tons. This is mainly because these countries lack the capacity for large-scale production thus, their energy usage is considerably low [97]. However even though the amount of emissions is significantly lower compared to the other economic development categories, Bangladesh emits 185% more $CO_2$ emissions in 2012–2021 compared to the average from 1995–2004. All top 10 $CO_2$ emitting countries in the least developed category emits at least 100% more $CO_2$ in 2012–2021 decade in comparison to the average from 1995–2004.

When comparing the results from Table 1 and S3 Appendix, it is evident that the top 10 $CO_2$-emitting countries in each category are almost the same in both periods. However, the emissions have increased significantly in all the categories except for developed category and a few countries from transitional economies. This shows that the countries responsible for the high levels of $CO_2$ have remained constant for the past 25 years, and the negative impact has been increasing over the years. The most significant difference can be observed in China. It averages 10290.07 million tons within 2012–2021 which is 165% more of China's $CO_2$ emissions from 1995–2004, the highest $CO_2$ emissions in that period. The next most considerable increment can be observed in India, with 944.87 million tons on average from 1995–2004 to an average of 2365.84 million tons between 2012–2021 which is an 150% increment over the two decades. US, Russia and Japan are in the top 5 $CO_2$-emitting countries, emitting 5255.338, 1664.01 and 1186.1 million tons, respectively. The increases suggest that these countries should focus on cleaner energy sources to reduce their excess carbon footprint to ensure that the effects of climate change and global warming would not be catastrophic in the long run.

According to Table 2 and S4 Appendix the REC as a percentage of TEC is constantly the highest in the least developed countries in both periods, compared to other economies. However, there was a clear decreasing trend in the REC as a percentage of TEC moved towards 2012–2021, except in the developed economies. Although developed nations consume more RE, the REC as a percentage of TEC is lower than the least developed economies as the energy consumption from non-renewable energy sources are higher. However, the usage levels of RE have increased with time, demonstrating a dedication to sustainable energy practices.

Malta, Macao, United Kingdom, Belgium, Netherlands, Ireland, Cyprus, and Japan were the countries with developed economies with the lowest REC as a percentage of TEC for both periods (Table 2 and S4 Appendix). However, a significant rise for 2012–2021 was visible, specifically in Malta, Macao, United Kingdom and Belgium where their percentage increment is 19813%, 1035%, 794% and 545%. For the developing economies, Algeria, Bahamas, Iraq, Iran, Hong Kong, Saudi Arabia, and Singapore are highlighted as countries with the lowest REC as a percentage of TEC for both periods (S4 Appendix). Saudi Arabia consistently recorded the lowest value for the REC as a percentage of TEC in both periods. It is the country with the lowest average REC as a percentage of TEC in the world in 2012–2021 period although it showed a slight increase for the period of 2012–2021. For transitional economies, most of the countries show an increase in the REC as a percentage of TEC while Kazakhstan, Russia, Kyrgyzstan and Georgia show a 18%, 9%, 23& and 44% decrease respectively. Among the least developed economies, Mauritius shows the lowest average REC as a percentage of TEC between 2012–2021 period with a significant decrease from 26.18% to 8.89% from 1995–2012 to 2012–2021. Similar drops were observed in all 10 countries with the lowest average REC in least developed countries. However, developing and least-developed nations still lag in implementing RES. These tables demonstrate the necessity of expediting the efforts and international collaboration to encourage the use of RE worldwide, especially in areas where consumption is still significantly low. Promoting and investing in RE systems is essential to mitigate climate change and create a more sustainable future. Increased utilisation of RES would significantly reduce

**Table 3. Results from the specification test for each economic development category.**

| Country Category | Specification Tests | | |
|---|---|---|---|
| | **F Test** | **Hausman Test** | **Breusch Pagan LM Test** |
| | $H_0$: POLS | $H_0$: REM | $H_0$: POLS |
| | $H_1$: FEM | $H_1$: FEM | $H_1$: REM |
| All Countries | 259.75*** | 1.57 | 39623.56*** |
| Developed | 5302.39*** | 0.89 | 13151*** |
| Developing | 172.52*** | 11.16*** | 15875*** |
| Economies in Transition | 4746.39*** | 2.55 | 4073.47*** |
| Least Developed | 117.73*** | 7.59* | 5527.88*** |

Note: ***, **, * represent 1%, 5% and 10% significance levels, respectively.

Source: Authors' calculations based on data from World Bank and Our World in Data.

pollution, environmental impact, and emissions contributing to the overall mitigation of climate change [21, 63]. The results of the specification tests carried out for each economic development category are presented in Table 3.

According to the results from the F test, $H_0$ of all five economic development categories were rejected indicating that the FEM model is more suitable. Moving on to the Hausman test, the FEM model is suggested for the developing and least developed economies as the results are significant at the 1% and 10% levels, respectively. The outcome of the Breusch Pagan LM test state that the REM model should be employed for all five categories. Considering the findings, the FEM model was utilised for the developing and least developed economies whilst the REM model was employed for all countries, developed and economies in transition. S5 Appendix presents the results for the fixed and random effect models for each economic development category. S6 Appendix provides the simple linear regression results and the multiple non-linear regression (maximum of 4 orders) analysis run for all the individual countries. Based on these findings, the 12 highest $CO_2$ emitting countries were selected, and the regression results are presented in S7 Appendix and are visualised through a line graph, as shown in (Fig 2).

In BRICS countries including China, the country with the highest $CO_2$ emissions, together with India and South Africa, a decreasing trend can be observed [41]. When REC increases by 1%, the $CO_2$ emissions decrease by approximately 341 million tons, 76 million tons and 10 million tons, respectively. In contrast to fossil fuel-based generators, RES, such as solar and wind power, provide electricity without releasing GHG while in operation. The total impact of emissions may be decreased due to this change in energy sources. The proportion of power generated from RES rises as more RE generation comes onboard, reducing the dependence on fossil fuels and lowering $CO_2$ emissions.

Analysing the findings for the United States, the second-highest $CO_2$ emitting nation, a significant negative trend can be observed between the variables where $CO_2$ emissions decline by 131 million tons per 1% rise in REC. RE systems, particularly solar panels and wind turbines may be installed at several scales. Since RE generation is distributed, less long-distance electricity transmission is required, and the impacts are more immediate [98], decreasing distribution and transmission costs. These reductions are connected to burning lesser fossil fuels, which lowers $CO_2$ emissions.

Further, the results for Germany, Italy and the United Kingdom also display a decreasing trend where a 1% increment in REC results in a decline in $CO_2$ emissions by approximately 14 million tons, 10 million tons and 21 million tons, respectively. A study based on European Union countries concludes a significant negative association between the variables for the

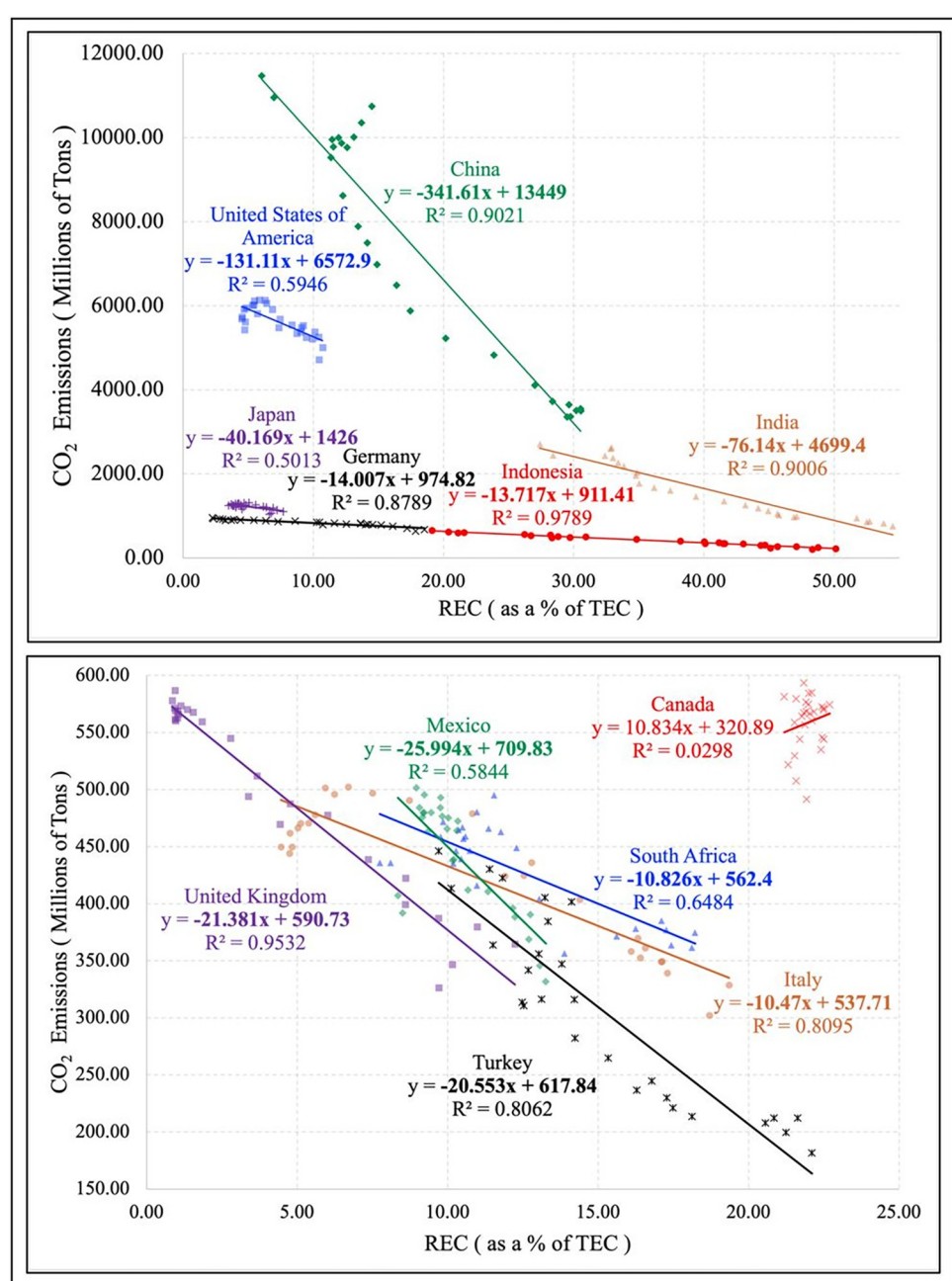

**Fig 2. Top 12 CO$_2$ emitting countries.** Source: Authors' illustration based on data from World Bank and Our World in Data.

United Kingdom [33], while the results for Germany and Italy are contradictory. A possible reason for this is the difference in the time frame of the data collected. These nations may use RE policies and established goals to raise the proportion of renewables in the energy mix. These policies offer incentives and support systems for the development of RE, resulting in lower CO$_2$ emissions.

Considering Mexico, a negative relationship between REC and CO$_2$ emissions can be detected where a 1% rise in REC leads to a 25.99 million tons decrease in CO$_2$ emissions.

Switching to RES additionally tackles regional air-quality concerns, particularly in densely populated areas, since RES emits lower particle emissions than coal-fired power plants. This ultimately results in a detrimental impact on $CO_2$ emissions.

Japan shows a similar relationship between REC and $CO_2$ emissions as well. According to the findings, a one per cent rise in REC would reduce 40.16 million tons of $CO_2$ emissions. Previous research on the decreasing trend between RE and $CO_2$ in Austria, Japan and New Zealand confirmed that REC lowers $CO_2$ emissions, mitigating climate change and environmental protection [55, 56]. Considering the nation's current state, the country would highly benefit economically and environmentally with increased investments in RES. Solar and wind power have become more affordable in recent years due to economies of scale and technological developments in the RE sector. RE has become increasingly enticing due to falling costs for utilities and consumers, increasing adoption, and lowering $CO_2$ emissions.

Similarly, Indonesia presents a similar relationship between the variables where $CO_2$ emissions decrease by approximately 13 million tons per one per cent increase in REC. Coal and oil have historically accounted for a large portion of the energy mix. To assist sustainable development and climate change mitigation, the amount of power produced by RES in Malaysia and Indonesia must be expanded, as REC directly contribute to reducing $CO_2$ emissions [37, 69]. These nations can gradually lessen its dependence on power generated from fossil fuels by boosting its use of RES, including solar, wind, and hydroelectric power. This moves toward alternatives to fossil fuels that reduce the $CO_2$ emissions linked to power generation.

Moreover, the findings for Turkey indicate a 20 million tons decline in $CO_2$ emissions with a one per cent increase in REC. Turkey's energy security can be improved by diversifying its energy mix, including renewable sources, lowering its dependency on imported fossil fuels and fostering domestic energy generation. Similarly, investing in diverse RES, promoting RES through incentives and infrastructure, and implementing policies that facilitate the transition to RES can be collectively done to propel Turkey towards successfully reducing carbon emissions.

In contrast, the analysis carried out for Canada presented a positive relationship between the two variables, where a one per cent rise in REC results in a 10.83 million tons increment in $CO_2$ emissions. The development of RES might be outpaced if the need for energy rises. When this occurs, energy sources based on fossil fuels may be employed to fulfil the increasing demand, resulting in more $CO_2$ emissions. Additionally, intermittent RES like wind and solar power depend on elements such as weather. There may be a need for alternative energy production facilities to guarantee a constant power supply. These backup systems may increase $CO_2$ emissions if they use fossil fuels.

## Conclusion

The significance of RE has been emphasised through this study, considering the energy crisis and the risks of continually increasing $CO_2$ emissions on the environment. Previous studies have concentrated on a single country or a few nations. In contrast, this study offers a more comprehensive analysis based on a substantial number of countries, covering a more longitudinal time range, utilising the most recent data and addressing the research gaps. Based on the results, only 12 nations with the most $CO_2$ emissions were chosen to study the relationship between $CO_2$ and RE due to the adverse environmental impacts.

The study's findings, which showed that almost all nations except Canada had a declining trend, highlighted the possibility that an increment in RES might reduce the global effect of emissions. Given the status of many nations, increasing their investment in RES would be highly advantageous on both an economic and environmental context as such sources restrict

the emissions of GHG. Further, the current analysis provides evidence that Canada shows a positive relationship between the variables. Investing in initiatives to enhance the usage of RES would lead to a substantial reduction in non-renewable energy consumption, resulting in a significant decrease in $CO_2$ emissions over the long term. In developed economies, investments in research and development are crucial for identifying methods to further utilise RES in climate mitigation. Concurrently, developing economies should prioritise increased investments in RE while minimising funds allocated to fossil fuels. Collaborations between governments of developing and developed economies are recommended to facilitate the sharing of financial resources and best practices for RE development [23, 43]. Furthermore, these findings also hold the potential to educate the general public on the pivotal role of RES in reducing $CO_2$ emissions. Increased awareness in these areas is expected to drive positive changes in consumer behaviour and inspire innovative initiatives to address the effects of climate change.

The significance that policy promotion plays in the growth of RE should be taken seriously by the government. Only through analysing RE policies, one of the clean energy resources, is it feasible to figure out sustainable development, which results from the consideration of the environment when guaranteeing economic growth.

## Policy implications

Transitioning towards RE is a complex exercise. It requires comprehensive policy measures covering economic, social, and technological factors which is more challenging, particularly for countries heavily dependent on fossil fuels for economic activities and energy security. Further, a collective effort of the nations to reduce $CO_2$ emissions would significantly impact global climate mitigation. This study suggests the following recommendations for countries with the highest $CO_2$ emissions to embrace a successful transition towards RES, which helps to mitigate $CO_2$ emissions and combat climate change.

Developed countries with a strong economic background such as Germany, Italy, Japan, United Kingdom, and Unites States must invest in research and development initiatives focusing on advanced RET and their integration into the energy grid, which will help overcome technological barriers and reduce the initial costs associated with RE implementation. The research and development initiatives of the developed countries with less REC as a percentage of TEC such as Germany, Japan, United Kingdom, and United States must focus on developing and modernising energy infrastructure to accommodate a higher share of RES. These expansions must accommodate smart grid technology implementation and energy storage solutions to ensure a resilient and reliable energy supply. However, developed economies which shows a positive relationship between REC and $CO_2$ emissions such as Canada, must invest in alternative backup systems for intermittent renewable sources. Such efforts can prevent increased $CO_2$ emissions and must maintain a balanced energy mix integrating RE and fossil fuel sources to ensure $CO_2$ emission reductions are maintained. Furthermore, these developed countries can consider RE exports to neighbouring countries which will strengthen the economic relationships and support to reduce emissions. Also, governments must invest in research and development of RET enhanced energy production and improved energy storage, distribution, and grid management which facilitates energy storage systems to store excess energy during peak production and release it during high demand. Electrochemical energy storage technologies are one of the most suitable applications for RE due to their higher efficiency and flexible designs. However, RE storage faces challenges in scalability and cost-effectiveness; thus, it is ideal for developed economies. Also, floating solar panels on reservoirs and offshore wind farms can be recommended for countries with limited land resources such as Italy and Japan [99]. Moreover, governments must highly encourage the adoption of electric

vehicles and support the rapid development of electric vehicle charging infrastructure to reduce $CO_2$ emissions from the transportation sector. Furthermore, developed economies can consider satellite observations of $CO_2$ emissions to get more accurate and timely data.

For the developing countries such as China, India, Indonesia, Mexico, South Africa, and Turkey, where increased REC has caused a substantial reduction in $CO_2$ emissions, the investments and resource allocation towards RE projects should be increased. They can also gradually minimize the investments towards fossil fuel industries to accelerate RES adoption to be economically competitive. This can enhance energy security by improving energy independence and reducing the risk of being vulnerable to price fluctuations in global fossil fuel markets. Investing in advanced RES innovation will help reduce the costs associated with RE deployment and improve the efficiency and scalability of RES. Further, advanced Carbon Capture, Utilization and Storage technologies should be implemented on larger scales among countries such as China and India. Countries with rich ecosystems such as China, India, Mexico and Indonesia can enforce policies promoting afforestation and reforestation as the terrestrial and marine ecosystems act as $CO_2$ reservoirs which could increase carbon sequestration and mitigate climate change. Moreover, developing economies with modest REC as a percentage of TEC such as Mexico and Turkey must further diversify the energy mix by including more RES, to reduce dependence on fossil fuels. The governments and local industrial experts of above developing countries must form international collaborations and partnerships with governments with developed economies, and foreign industrial experts to share best practices, technological knowledge, and financial resources for RE development.

Moreover, all the governments of the above mentioned 12 highest $CO_2$ emitting countries must establish clear objectives and timelines for the implementation of RES. These objectives and timelines must be developed as a long-term energy transition strategy to avoid the usage of fossil fuels and maximise REC. These strategies should be periodically reviewed and methodically adjusted based on evolving technological, economic, and environmental considerations. Finally, supportive policy frameworks that facilitate the transition to RE should be implemented. Moreover, these governments must endeavour to attract private and public investment by providing clear regulatory frameworks, financial incentives for RE projects, and launching public awareness campaigns to educate citizens about the benefits of RE and the importance of carbon emission reduction. Building public support can drive demand for RES and encourage more outstanding political commitment. Moreover, implementation of carbon pricing mechanisms such as carbon taxes or to provide economic incentives for reducing $CO_2$ emissions must be enabled. Also, governments must invest in re-training programs to transition employees involved in fossil fuel industries to RE sectors to build a capable workforce that can support the RE transition and be skilled in designing, installing, operating, and maintaining RE systems.

Finally, all countries must make significant and consistent efforts to mitigate $CO_2$ emissions, combat climate change, and foster sustainable economic growth to ensure a brighter future by diligently implementing strategies and embracing comprehensive approaches to transit into using RES for energy production.

## Managerial implications

The top 12 $CO_2$ emitting countries can gain many insightful recommendations from this study. Apart from Canada, where there's a positive relationship from REC to $CO_2$, all 11 other countries can highly promote RE related products in their most revenue generating industries. Countries such as India and China can focus on implementing RE projects to their manufacturing and production industry. Initiatives such as these projects will lead to

sustainable revenue generating industries while generating opportunities for the community in the form of project managers, project management teams etc. Project managers will be further inspired to develop strategic plans to initiate high value collaborations [29]. With the involvement of the government, investors and other external stakeholders, renewable energy projects can bring an immense amount of value to these countries in their journey to achieving sustainable development.

## Limitations

As this paper focused on identifying the relationship from RE to $CO_2$ emissions over 27 years for 138 nations, the primary limitation was the absence of the source of RE during the study period. Therefore, it would be challenging to explore more in future studies about solar power, bioenergy, and geothermal power affect economic growth. Moreover, future studies should evaluate the connection between RE, particularly wind and solar energy, and additional GHG and ecological footprint in addition to $CO_2$ emissions.

## Supporting information

**S1 Appendix. Data file.**
(XLSX)

**S2 Appendix. Descriptive statistics for renewable energy consumption (% of final energy consumption) and $CO_2$ emissions (millions of tons).**
(DOCX)

**S3 Appendix. Difference of averages of $CO_2$ emissions from 1995–2004 and 2012–2021.**
(DOCX)

**S4 Appendix. Difference of averages of renewable energy consumption from 1995–2004 and 2012–2021.**
(DOCX)

**S5 Appendix. Results for fixed and random effect models for each economic development category.**
(DOCX)

**S6 Appendix. Regression results for individual countries.**
(DOCX)

**S7 Appendix. Countries with the highest significant positive coefficients.**
(DOCX)

## Author Contributions

**Conceptualization:** Keshani Attanayake, Isuru Wickramage, Udul Samarasinghe, Yasangi Ranmini, Ruwan Jayathilaka.

**Data curation:** Keshani Attanayake, Udul Samarasinghe.

**Formal analysis:** Keshani Attanayake, Isuru Wickramage, Udul Samarasinghe, Yasangi Ranmini, Sandali Ehalapitiya, Ruwan Jayathilaka.

**Investigation:** Ruwan Jayathilaka.

**Methodology:** Keshani Attanayake, Udul Samarasinghe, Ruwan Jayathilaka.

**Software:** Keshani Attanayake, Isuru Wickramage, Yasangi Ranmini.

**Supervision:** Ruwan Jayathilaka, Shanta Yapa.

**Validation:** Keshani Attanayake, Udul Samarasinghe, Ruwan Jayathilaka.

**Visualization:** Udul Samarasinghe, Ruwan Jayathilaka.

**Writing – original draft:** Keshani Attanayake, Isuru Wickramage, Udul Samarasinghe, Yasangi Ranmini, Sandali Ehalapitiya, Ruwan Jayathilaka.

**Writing – review & editing:** Sandali Ehalapitiya, Ruwan Jayathilaka, Shanta Yapa.

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
