## [Decision Letter · Decision Letter 0]

5 Oct 2023

PONE-D-23-27314Renewable Energy as a Solution to Climate Change: Insights from a Comprehensive Study Across NationsPLOS ONE

Dear Dr. Jayathilaka,

Thank you for submitting your manuscript to PLOS ONE. After careful consideration, we feel that it has merit but does not fully meet PLOS ONE’s publication criteria as it currently stands. Therefore, we invite you to submit a revised version of the manuscript that addresses the points raised during the review process.

Please revise your paper according to reviewers' reports and mark into color all changes you will make into the manuscript. Also prepare a response letter for reviewers.

We look forward to receiving your revised manuscript.

Kind regards,

Magdalena Radulescu

Academic Editor

PLOS ONE

Journal Requirements:

2. We note that Figures 2(A), 2(B) and 3(A), 3(B) in your submission contain map images which may be copyrighted. All PLOS content is published under the Creative Commons Attribution License (CC BY 4.0), which means that the manuscript, images, and Supporting Information files will be freely available online, and any third party is permitted to access, download, copy, distribute, and use these materials in any way, even commercially, with proper attribution. For these reasons, we cannot publish previously copyrighted maps or satellite images created using proprietary data, such as Google software (Google Maps, Street View, and Earth). For more information, see our copyright guidelines: http://journals.plos.org/plosone/s/licenses-and-copyright.

We require you to either present written permission from the copyright holder to publish these figures specifically under the CC BY 4.0 license, or (2) remove the figures from your submission:

a. You may seek permission from the original copyright holder of Figures 2(A), 2(B) and 3(A), 3(B) to publish the content specifically under the CC BY 4.0 license.  

I request permission for the open-access journal PLOS ONE to publish XXX under the Creative Commons Attribution License (CCAL) CC BY 4.0 (http://creativecommons.org/licenses/by/4.0/). Please be aware that this license allows unrestricted use and distribution, even commercially, by third parties. Please reply and provide explicit written permission to publish XXX under a CC BY license and complete the attached form.

In the figure caption of the copyrighted figure, please include the following text: “Reprinted from [ref] under a CC BY license, with permission from [name of publisher], original copyright [original copyright year]."

Additional Editor Comments:

Please revise your paper according to reviewers' reports and mark into color all changes you will make into the manuscript. Also prepare a response letter for reviewers.

Comments from PLOS Editorial Office: We note that one or more reviewers has recommended that you cite specific previously published works. As always, we recommend that you please review and evaluate the requested works to determine whether they are relevant and should be cited. It is not a requirement to cite these works. We appreciate your attention to this request.

Reviewers' comments:

Reviewer's Responses to Questions

**Comments to the Author**

1. Is the manuscript technically sound, and do the data support the conclusions?

Reviewer #1: No

Reviewer #2: Yes

2. Has the statistical analysis been performed appropriately and rigorously? 

Reviewer #1: No

Reviewer #2: Yes

3. Have the authors made all data underlying the findings in their manuscript fully available?

Reviewer #1: Yes

Reviewer #2: Yes

4. Is the manuscript presented in an intelligible fashion and written in standard English?

Reviewer #1: Yes

Reviewer #2: Yes

5. Review Comments to the Author

Reviewer #1: Thank you very much for inviting me to assess the above-mentioned manuscript submitted to Plos One. From the research topic, this paper attempts to examine "Renewable Energy as a Solution to Climate Change: Insights from a Comprehensive Study Across Nations" approaches.

I suggest authors to rewrite the abstract to make it more constructive. Abstract should have at least one sentence per each: context and background, motivation, hypothesis, methods, results, conclusions.

Intro & Literature: i) The introduction part of the study needs improvement and story flow and the authors need to give proper contributions to their study.

ii) There is a need to do a more rigorous and systematic literature review. Give the touch of more environmental theory

See and kindly add following paper to the references https://doi.org/10.1080/15567249.2016.1263251;
https://doi.org/10.1007/s13132-011-0075-2;
https://doi.org/10.1007/s12667-010-0018-1;
https://doi.org/10.1007/s11356-019-04514-6;
https://doi.org/10.1007/s11356-021-16720-2;
https://doi.org/10.1007/s11356-021-12993-9;
https://doi.org/10.1007/s11356-023-26675-1;
https://doi.org/10.1007/s11356-023-28977-w;
https://doi.org/10.1108/IJCHM-10-2022-1201;
https://doi.org/10.1007/s11356-023-29020-8

Take out the footnotes from text

Policy recs. are a bit vague and not convincing to the reader.

It would be appropriate to indicate future research directions and limitations of this at the end of the conclusion section just before references. Need clear future recommendation/implementation in the context of uncertainty

https://doi.org/10.1007/s00477-023-02452-x;
https://doi.org/10.1177/1354816619888346

Reviewer #2: 1)Major findings or trends found as a result of the study.

- A summary of your interpretations and conclusions.

2. All abbreviations must be introduced at first mention. Consider the abstract independently. Only recheck it, please.

3. The Introduction should have valuable aspects such as motivation, problem, challenges, the significance of the study…etc. All these are disappearing. Therefore, the introduction needs to clarify the (1) motivation, (2) challenges, (3) contribution, (4) objectives, and (5) significance/implication of this systematic. All the information (should be) presented in sequence idea.

4. I can see that the presented tables lack analysis. The authors should add a brief description and provide an analysis of the results in these tables to be clear for the readers.

5. What are the limitations of the presented study? I missed this part.

6. Double-check for grammar, typos, and abbreviations.

7. Improve the English language by profession. Improve Your Style in Academic Writing

8. A Lot of studies have been published in the literature. What is new in this work? The authors should prove the novelty and originality of the presented work.

9. Rewrite the Conclusion and consider the following comments:

- Highlight your analysis and reflect only the essential points for the paper.

- Mention the benefits

- Mention the implication in the last of this section.

10. Remove the duplication between abstract and Conclusion.

6. PLOS authors have the option to publish the peer review history of their article (what does this mean?). If published, this will include your full peer review and any attached files.

Reviewer #1: No

Reviewer #2: **Yes: **Shabir Hussain Malik

---

## [Author Response · Author response to Decision Letter 0]

27 Oct 2023

Point by point response to editor and reviewers.

We would like to express our profound appreciation to the editor for giving us the opportunity to revise and resubmit the paper. We are also grateful to the reviewers for the constructive comments and suggestions made on our manuscript which we found very helpful in revising and improving our paper. Given below is a detailed description of how we have addressed each comment in the revised version of the paper.

Editor’s comment 1: Please revise your paper according to reviewers' reports and mark into color all changes you will make into the manuscript. Also prepare a response letter for reviewers.

Authors’ Response to Editor Comment 1: Thank you for your valuable feedback. We have thoroughly reviewed the reviewers' reports and incorporated the necessary revisions into the manuscript. In response to your request, we have marked all changes in colour within the document. Additionally, we have prepared a response letter to address each of the reviewers' comments. These responses are included in this document for your convenience.

Editor’s comment 2: We note that one or more reviewers has recommended that you cite specific previously published works. As always, we recommend that you please review and evaluate the requested works to determine whether they are relevant and should be cited. It is not a requirement to cite these works. We appreciate your attention to this request.

Authors’ Response to Editor Comment 2: Thank you for your comment. We have carefully reviewed the suggested articles and have incorporated the majority of them into the manuscript's references. These additions have been made because they are highly relevant to the focus area of our study. We appreciate your guidance in this regard and believe that these citations will enhance the quality and comprehensiveness of our work

Reviewer 1 comment 1: Thank you very much for inviting me to assess the above-mentioned manuscript submitted to Plos One. From the research topic, this paper attempts to examine "Renewable Energy as a Solution to Climate Change: Insights from a Comprehensive Study Across Nations" approaches.

Authors’ Response to Reviewer 1 comment 1: Thank you very much for your assessment of the manuscript titled 'Renewable Energy as a Solution to Climate Change: Insights from a Comprehensive Study Across Nations.' We appreciate your thoughtful comments and the time you've dedicated to reviewing our work. The authors are truly grateful for the feedback, which will undoubtedly contribute to enhancing the quality of the manuscript.

Reviewer 1 comment 2: I suggest authors rewrite the abstract to make it more constructive. Abstract should have at least one sentence per each: context and background, motivation, hypothesis, methods, results, conclusions.

Authors’ Response to Reviewer 1 comment 2: Thank you for your valuable feedback. We have revised the abstract to better align with the suggested structure, ensuring a more comprehensive and constructive summary of our research.

In the updated abstract (lines 24 - 35), we have provided a clear breakdown as follows:

Context and Background: “Without fundamentally altering how humans generate and utilise energy, there is no effective strategy to safeguard the environment”

Motivation: “The motivation behind this study was to analyse the effectiveness of renewable energy in addressing climate change, as it is one of the most pressing global issues.”

Hypothesis: “This study involved the analysis of panel data covering 138 nations over a 27 year period, from 1995 to 2021, making it the latest addition to the existing literature.”

Methods: “We examined the extent of the impact of renewable energy on carbon dioxide over time using panel, linear, and non-linear regression approaches.”

Results: “The results of our analysis, revealed that the majority of countries with the exception of Canada, exhibited a downward trend, underscoring the potential of increasing renewable energy consumption as an effective method to reduce carbon dioxide emissions and combat climate change.”

Conclusions: "Furthermore, to reduce emissions and combat climate change, it is advisable for nations with the highest carbon dioxide emissions to adopt and successfully transition to renewable energy sources.”

We believe these revisions will provide a clearer and more structured overview of our research. Once again, we appreciate your valuable input.

Reviewer 1 comment 3: Intro & Literature: i) The introduction part of the study needs improvement and story flow and the authors need to give proper contributions to their study.

ii) There is a need to do a more rigorous and systematic literature review. Give the touch of more environmental theory

See and kindly add following paper to the references https://doi.org/10.1080/15567249.2016.1263251;
https://doi.org/10.1007/s13132-011-0075-2;
https://doi.org/10.1007/s12667-010-0018-1;
https://doi.org/10.1007/s11356-019-04514-6;
https://doi.org/10.1007/s11356-021-16720-2;
https://doi.org/10.1007/s11356-021-12993-9;
https://doi.org/10.1007/s11356-023-26675-1;
https://doi.org/10.1007/s11356-023-28977-w;
https://doi.org/10.1108/IJCHM-10-2022-1201;
https://doi.org/10.1007/s11356-023-29020-8

Authors’ Response to Reviewer 1 comment 3: Thank you for your insightful feedback on the introduction and literature review of our manuscript. We have taken your suggestions seriously and made significant improvements to these sections.

We've enhanced the flow of the introduction to create a more cohesive and logical narrative. Additionally, we have clearly outlined the contributions of our study, which are now elaborated on in lines 89 – 112.

“This study contributes to the existing literature in several fronts. Firstly, it analyses each country separately and categorises them based on their level of economic development as developed, developing, least developed, and transitional economies. Despite extensive research on the relationship between CO2 emissions and RE of developed and developing economies, a significant gap in the literature exists regarding the context of least developed and transitional economies. Focusing on understudied areas, our research aims to contribute to this limited body of literature by addressing the context of least developed and transitional economies to provide valuable insights for policymakers. Secondly, panel regression, linear regression and non-linear regression techniques are used in this study, allowing for a more comprehensive analysis of both linear and non-linear relationships between CO2 emissions and RE and the level of the impact on the respective variables over a period. This is important in understanding how the relationship between CO2 emissions and RE has evolved across different countries and accurately identifying patterns and changes. Thirdly, this study visualises the top 12 CO2 emitting countries, and the relationship between REC and CO2 emissions in each country in a single graph allowing the users to identify the countries who negatively impacts the environment the most and how they can reduce those effects by transitioning to RES. Finally, this study represents the latest addition to the current literature, analysing panel data from 1995 to 2021 over 27 years covering 138 nations. While previous research has focused on individual countries or a limited number of countries, there need to be more comprehensive global studies to provide a thorough understanding of the worldwide relationship between CO2 emissions and RE for a significant period. Therefore, this study provides a broader analysis based on a more substantial number of countries considering a more comprehensive period using the latest available data to obtain a more accurate depiction of the global status of the relationship between two variables helping for more precise decision-making.”

We have strengthened our literature review by incorporating a substantial number of the suggested articles into our manuscript. These additions can be found in lines 50, 65, 157 and 174. 

These additional references not only enrich the theoretical framework but also provide a more comprehensive understanding of the topic, reinforcing the scholarly depth of our study.

“Thus, the adverse environmental impacts, as well as the factors which influence CO2 emissions have remained an important subject of worldwide discussions among the academic community for many years [7, 8].”

“Hence, the academic community has conducted multiple studies on achieving sustainability goals, carbon neutrality, alternative options for fossil fuels such as natural gases, RE and their impact towards environmental sustainability [12-16].”

“Moreover, multiple studies based on Organization for Economic Cooperation and Development (OECD) countries concluded that REC and investment in RE reduces CO2 emissions while helping to achieve environmental sustainability [16, 33, 34].”

“Further, multiple studies about CO2 emissions and RE in the United States collectively concluded that despite the rising public acceptance of the fact that burning of fossil fuels is a major reason for climate change and the usage of RE can mitigate climate change, the levels of REC of the United States cannot significantly contribute to CO2 emissions reduction yet [38-43].”

Reviewer 1 comment 4: Take out the footnotes from text.

Authors’ Response to Reviewer 1 comment 4: Thank you for your feedback. Footnotes are only present on the title page to list the affiliations of each author, in accordance with the PLOS ONE journal guidelines. No other footnotes are used in the text. You can refer to the provided template for reference as below.

https://storage.googleapis.com/plos-published-prod/3fac/PLOS%20Affiliations%20Formatting%20Guidelines.pdf?X-Goog-Algorithm=GOOG4-RSA-SHA256&X-Goog-Credential=wombat-sa%40plos-prod.iam.gserviceaccount.com%2F20231013%2Fauto%2Fstorage%2Fgoog4_request&X-Goog-Date=20231013T055223Z&X-Goog-Expires=86400&X-Goog-SignedHeaders=host&X-Goog-Signature=705d17d35a38e1d4f8d1b2e81bd70814fa0dcc65a5344eeea1216daec8dda71638551d45cc8692306840241f7d5f7bff45a1026009a89107336696d54b0610ee34ab059be3131cd9b8ab70a6eb9ffe65fea0eda81eab5f1f33ca752ec79779eb2c526e0bccfa5440ff62104cf2ad3d06066a88536a6b2d6de2d8835435ea9e6d91ed3fd6764af2c08a785d400b729ae06ef7e8f2e62273ca8fe5d13e0d8e19118e6dbda2eb4d29b902ef9c7fee211f58c24b1f661fc7ad2089d3d7a285973c8602290ca129f1536386cefcdc47313969c1e8bf17ba10452a590b8a9c7d6458b84887955627e3068f216685ea93ecf8edf14972a7df5a8e544966a5d9ff0db7e6

Reviewer 1 comment 5: Policy recs. are a bit vague and not convincing to the reader.

Authors’ Response to Reviewer 1 comment 5: Thank you for your valuable input, and we appreciate your attention to the policy recommendations. We have taken your feedback seriously and have made significant improvements to the policy recommendations section.

The policy recommendations have been updated to provide more specific and convincing guidance for each country, considering all 12 of the highest CO2-emitting nations. These revisions aim to offer a more robust and actionable set of recommendations that address the challenges and opportunities presented by our study.

We believe that these enhancements have made our policy recommendations more concrete and persuasive, and we are grateful for your insights in this regard and these changes can be found between lines 467 -537.

“Transitioning towards RE is a complex exercise. It requires comprehensive policy measures covering economic, social, and technological factors which is more challenging, particularly for countries heavily dependent on fossil fuels for economic activities and energy security. Further, a collective effort of the nations to reduce CO2 emissions would significantly impact global climate mitigation. This study suggests the following recommendations for countries with the highest CO2 emissions to embrace a successful transition towards RES, which helps to mitigate CO2 emissions and combat climate change. 

Developed countries with a strong economic background such as Germany, Italy, Japan, United Kingdom, and Unites States must invest in research and development initiatives focusing on advanced RET and their integration into the energy grid, which will help overcome technological barriers and reduce the initial costs associated with RE implementation. The research and development initiatives of the developed countries with less REC as a percentage of TEC such as Germany, Japan, United Kingdom, and United States must focus on developing and modernising energy infrastructure to accommodate a higher share of RES. These expansions must accommodate smart grid technology implementation and energy storage solutions to ensure a resilient and reliable energy supply. However, developed economies which shows a positive relationship between REC and CO2 emissions such as Canada, must invest in alternative backup systems for intermittent renewable sources. Such efforts can prevent increased CO2 emissions and must maintain a balanced energy mix integrating RE and fossil fuel sources to ensure CO2 emission reductions are maintained. Furthermore, these developed countries can consider RE exports to neighbouring countries which will strengthen the economic relationships and support to reduce emissions. Also, governments must invest in research and development of RET enhanced energy production and improved energy storage, distribution, and grid management which facilitates energy storage systems to store excess energy during peak production and release it during high demand. Electrochemical energy storage technologies are one of the most suitable applications for RE due to their higher efficiency and flexible designs. However, RE storage faces challenges in scalability and cost-effectiveness; thus, it is ideal for developed economies. Also, floating solar panels on reservoirs and offshore wind farms can be recommended for countries with limited land resources such as Italy and Japan [91]. Moreover, governments must highly encourage the adoption of electric vehicles and support the rapid development of electric vehicle charging infrastructure to reduce CO2 emissions from the transportation sector. Furthermore, developed economies can consider satellite observations of CO2 emissions to get more accurate and timely data. 

For the developing countries such as China, India, Indonesia, Mexico, South Africa, and Turkey, where increased REC has caused a substantial reduction in CO2 emissions, the investments and resource allocation towards RE projects should be increased. They can also gradually minimize the investments towards fossil fuel industries to accelerate RES adoption to be economically competitive. This can enhance energy security by improving energy independence and reducing the risk of being vulnerable to price fluctuations in global fossil fuel markets. Investing in advanced RES innovation will help reduce the costs associated with RE deployment and improve the efficiency and scalability of RES. Further, advanced Carbon Capture, Utilization and Storage technologies should be implemented on larger scales among countries such as China and India. Countries with rich ecosystems such as China, India, Mexico and Indonesia can enforce policies promoting afforestation and reforestation as the terrestrial and marine ecosystems act as CO2 reservoirs which could increase carbon sequestration and mitigate climate change. Moreover, developing economies with modest REC as a percentage of TEC such as Mexico and Turkey must further diversify the energy mix by including more RES, to reduce dependence on fossil fuels. The governments and local industrial experts of above developing countries must form international collaborations and partnerships with governments with developed economies, and foreign industrial experts to share best practices, technological knowledge, and financial resources for RE development.

Moreover, all the governments of the above mentioned 12 highest CO2 emitting countries must establish clear objectives and timelines for the implementation of RES. These objectives and timelines must be developed as a

---

## [Decision Letter · Decision Letter 1]

21 Dec 2023

PONE-D-23-27314R1Renewable Energy as a Solution to Climate Change: Insights from a Comprehensive Study Across NationsPLOS ONE

Dear Dr. Jayathilaka,

Thank you for submitting your manuscript to PLOS ONE. After careful consideration, we feel that it has merit but does not fully meet PLOS ONE’s publication criteria as it currently stands. Therefore, we invite you to submit a revised version of the manuscript that addresses the points raised during the review process. 

We look forward to receiving your revised manuscript.

Kind regards,

Magdalena Radulescu

Academic Editor

PLOS ONE

Journal Requirements:

Additional Editor Comments:

*Comments from Journal: One or more of the reviewers has recommended that you cite specific previously published works. Members of the editorial team have determined that the works referenced are not directly related to the submitted manuscript. As such, please note that it is not necessary or expected to cite the works requested by the reviewer. *

Reviewers' comments:

Reviewer's Responses to Questions

**Comments to the Author**

1. If the authors have adequately addressed your comments raised in a previous round of review and you feel that this manuscript is now acceptable for publication, you may indicate that here to bypass the “Comments to the Author” section, enter your conflict of interest statement in the “Confidential to Editor” section, and submit your "Accept" recommendation.

Reviewer #1: All comments have been addressed

Reviewer #3: (No Response)

2. Is the manuscript technically sound, and do the data support the conclusions?

Reviewer #1: Yes

Reviewer #3: Partly

3. Has the statistical analysis been performed appropriately and rigorously? 

Reviewer #1: Yes

Reviewer #3: Yes

4. Have the authors made all data underlying the findings in their manuscript fully available?

Reviewer #1: Yes

Reviewer #3: Yes

5. Is the manuscript presented in an intelligible fashion and written in standard English?

Reviewer #1: Yes

Reviewer #3: Yes

6. Review Comments to the Author

Reviewer #1: Thank you for giving me the opportunity to read your paper. The paper “PONE-D-23-27314R1” is interesting for journal readers. My oinion is "accept"

Reviewer #3: Upon thorough review, I have determined that the article need further enhancement in some areas in order to increase its overall quality:

- Renewable energy serves as a viable way to mitigate climate change. To substantiate this claim, it is necessary to include further information in the last paragraph describing how this research will make a distinctive contribution.

- The current literature on renewable energy consumption and climate change lacks more recent information. The authors could additionally focus on environmental deterioration and carbon emissions. Here, I am providing some recommended research to enhance this section:

(1)-https://doi.org/10.1016/j.gr.2023.11.002 (2)-https://doi.org/10.1016/j.scitotenv.2023.168027 (3)-https://doi.org/10.1016/j.gr.2023.08.019 (4)- https://doi.org/10.1177/21582440211060829 (5)- https://doi.org/10.3390/su15129413 (6)-https://doi.org/10.3390/su15075916 (7)-https://doi.org/10.1007/s11356-023-25316-x (8)-https://doi.org/10.3390/en15197180 (9)-https://doi.org/10.1007/s11356-022-22775-6 (10)-https://doi.org/10.1007/s11356-022-19317-5 (11)- https://doi.org/10.1007/s11356-021-15481-2

- Figure quality has to be improved further.

- Include some lines on the study's practical implications in the conclusion section.

7. PLOS authors have the option to publish the peer review history of their article (what does this mean?). If published, this will include your full peer review and any attached files.

Reviewer #1: No

Reviewer #3: No

---

## [Author Response · Author response to Decision Letter 1]

29 Dec 2023

Authors Response to Editor and Reviewers Comments 

Dear editor and reviewers, 

We appreciate the feedback provided on our manuscript to improve the overall quality of our work. Necessary changes have been made to the manuscript taking into account the comments provided. Your input has been incredibly helpful in revising and improving the quality of our work.

Please note that the line numbers referred to in this document are aligned with the revised manuscript which includes track changes.

Thank you once again for your valuable feedback.

Journal Requirements: Please review your reference list to ensure that it is complete and correct. If you have cited papers that have been retracted, please include the rationale for doing so in the manuscript text, or remove these references and replace them with relevant current references. Any changes to the reference list should be mentioned in the rebuttal letter that accompanies your revised manuscript. If you need to cite a retracted article, indicate the article’s retracted status in the References list and also include a citation and full reference for the retraction notice.

Authors’ Response to Journal Requirements: Thank you for your thoughtful review and constructive comments. We appreciate your diligence in ensuring the accuracy and completeness of our reference list.

Upon your suggestion, we meticulously reviewed our reference list to confirm its completeness and accuracy. We confirm that no retracted papers have been cited in the manuscript. In response to your guidance, we have made additional improvements to the reference list by incorporating eight recent references. The revised list now comprises a total of 98 references, up from the initial count of 90.

We would like to emphasize that every effort has been made to adhere to your recommendations, and we believe the reference list now aligns with the highest standards of scholarly accuracy.

Should you have any further concerns or require additional clarification, please do not hesitate to let us know. We look forward to your feedback on the revised manuscript.

Thank you for your time and valuable insights.

Editor’s comment: One or more of the reviewers has recommended that you cite specific previously published works. Members of the editorial team have determined that the works referenced are not directly related to the submitted manuscript. As such, please note that it is not necessary or expected to cite the works requested by the reviewer.

Authors’ Response to Editor Comment: Thank you for providing us with the feedback regarding the recommendations from the reviewers. We appreciate the editorial team's clarification on the relevance of certain previously suggested citations.

Upon careful consideration, we have reviewed the 11 articles recommended by the reviewer. After a thorough assessment, we have identified that 8 of these articles align closely with the scope of our manuscript and contribute substantively to our study. Therefore, we have incorporated these relevant works into our revised manuscript.

We believe that these additions enhance the overall quality and depth of our work without deviating from the focus of our research. We trust that this decision aligns with the expectations of both the reviewers and the editorial team.

If you have any further concerns or require additional information, please feel free to let us know. We are committed to ensuring that our manuscript meets the highest standards of scholarly rigor.

Reviewer 1 comment 1: Thank you for giving me the opportunity to read your paper. The paper “PONE-D-23-27314R1” is interesting for journal readers. My opinion is "accept"

Authors’ Response to Reviewer 1 comment 1: We sincerely appreciate your time and effort in reviewing our manuscript, and we are thrilled to learn that you find our work interesting for journal readers. Your positive feedback is invaluable to us.

We have duly noted your recommendation for acceptance. If there are any specific revisions or suggestions you would like us to address before the final submission, please do not hesitate to let us know. We are committed to ensuring that our paper meets the highest standards and aligns with the expectations of the journal.

Once again, thank you for your thoughtful review and support.

Reviewer 3 comment 1: Renewable energy serves as a viable way to mitigate climate change. To substantiate this claim, it is necessary to include further information in the last paragraph describing how this research will make a distinctive contribution.

Authors’ Response to Reviewer 3 comment 1: Thank you for your insightful comment. We have carefully incorporated the requested information into the manuscript, specifically in lines 347–351, 470–473, and 482–486.

In lines 347-351, we have added the following text to the last paragraph:

“Promoting and investing in RE systems is essential to mitigate climate change and create a more sustainable future. Increased utilisation of RES would significantly reduce pollution, environmental impact, and emissions contributing to the overall mitigation of climate change [21, 62].”

Additionally, in lines 470–473 and 482–486, the following statements have been included:

“Investing in initiatives to enhance the usage of RES would lead to a substantial reduction in non-renewable energy consumption, resulting in a significant decrease in CO2 emissions over the long term.”

“Furthermore, these findings also hold the potential to educate the general public on the pivotal role of RES in reducing CO2 emissions. Increased awareness in these areas is expected to drive positive changes in consumer behaviour and inspire innovative initiatives to address the effects of climate change.”

We believe that these additions strengthen the distinctive contribution of our research to the field.

Reviewer 3 comment 2: The current literature on renewable energy consumption and climate change lacks more recent information. The authors could additionally focus on environmental deterioration and carbon emissions. Here, I am providing some recommended research to enhance this section:

(1)-https://doi.org/10.1016/j.gr.2023.11.002 (2)-https://doi.org/10.1016/j.scitotenv.2023.168027 (3)-https://doi.org/10.1016/j.gr.2023.08.019 (4)- https://doi.org/10.1177/21582440211060829 (5)- https://doi.org/10.3390/su15129413 (6)-https://doi.org/10.3390/su15075916 (7)-https://doi.org/10.1007/s11356-023-25316-x (8)-https://doi.org/10.3390/en15197180 (9)-https://doi.org/10.1007/s11356-022-22775-6 (10)-https://doi.org/10.1007/s11356-022-19317-5 (11)- https://doi.org/10.1007/s11356-021-15481-2

Authors’ Response to Reviewer 3 comment 2: Thank you for providing valuable suggestions to enhance the literature on renewable energy consumption and climate change in our manuscript. We appreciate your effort in identifying relevant research articles. The following changes have been made to our manuscript in response to your recommendations:

Eight out of the eleven mentioned articles have been added to our literature, as they align closely with the variables addressed in our study. Our initial reference count of 90 has now increased to 98. The specific changes can be found in the manuscript at the following locations: lines 70-72, 73-75,168-171, 189-193, 200-204, and 347-351.

For instance

“Hence, minimising the usage of fossil fuel and optimising the use of RE would be the optimal option to attain carbon neutrality and sustainably fulfil the energy needs of humans as it can reduce CO2 emissions [18-20].”

“Transitioning to RE for energy production is essential as it helps reduce carbon emissions, mitigate climate change, enhance environmental sustainability, and foster long-term socio-economic benefits such as sustainable economic growth. [21-23]”

“Studies based on developed economies namely Canada, France, Japan, Netherlands, Spain, Sweden, Switzerland, United Kingdom, and the United States have concluded that RE reduces CO2 emissions and the use of RE is beneficial for environmental protection [41, 42].”

“Despite being recognised as the world’s largest CO2 emitter, China must focus on reducing its CO2 emissions by utilising RE. Multiple studies have concluded that the adoption of RE has a significant negative impact on CO2 emissions in China, contributing to the mitigating climate change [59-62].”

“Similar studies in China, Pakistan, Malaysia, and Indonesia have revealed that increasing and actively funding the proportion of power generated from RES is crucial to support sustainable development, climate change mitigation, and environmental protection [36, 62, 68, 69].”

“Promoting and investing in RE systems is essential to mitigate climate change and create a more sustainable future. Increased utilisation of RES would significantly reduce pollution, environmental impact, and emissions contributing to the overall mitigation of climate change [21, 62].”

We believe these additions strengthen the foundation of our study and contribute significantly to the existing literature. Once again, thank you for your insightful feedback and guidance.

Reviewer 3 comment 3: Figure quality has to be improved further.

Authors’ Response to Reviewer 3 comment 3: Thank you for your feedback. We acknowledge your comment regarding the figure quality. In response, we have made improvements to the figure by enhancing the colours and adjusting the font size to enhance legibility.

The revised version of the figure has been carefully incorporated into the manuscript. We believe these enhancements address the concerns raised and contribute to the overall clarity of the visual representation.

We appreciate your time and valuable insights.

Reviewer 3 comment 4: Include some lines on the study's practical implications in the conclusion section.

Authors’ Response to Reviewer 3 comment 4: Thank you for your valuable feedback. We have incorporated lines on the study's practical implications into the conclusion section as suggested. Please find the relevant additions in lines 474–481 and 482–486:

“In developed economies, investments in research and development are crucial for identifying methods to further utilise RES in climate mitigation. Concurrently, developing economies should prioritise increased investments in RE while minimising funds allocated to fossil fuels. Collaborations between governments of developing and developed economies are recommended to facilitate the sharing of financial resources and best practices for RE development [23, 42].”

“Furthermore, these findings also hold the potential to educate the general public on the pivotal role of RES in reducing CO2 emissions. Increased awareness in these areas is expected to drive positive changes in consumer behaviour and inspire innovative initiatives to address the effects of climate change.”

We believe these additions strengthen the practical implications of our study. Thank you for your insightful feedback.

---

## [Decision Letter · Decision Letter 2]

17 Jan 2024

PONE-D-23-27314R2Renewable Energy as a Solution to Climate Change: Insights from a Comprehensive Study Across NationsPLOS ONE

Dear Dr. Jayathilaka,

Thank you for submitting your manuscript to PLOS ONE. After careful consideration, we feel that it has merit but does not fully meet PLOS ONE’s publication criteria as it currently stands. Therefore, we invite you to submit a revised version of the manuscript that addresses the points raised during the review process.

Make a response letter and highlight in color all your addings into the revised manuscript.

We look forward to receiving your revised manuscript.

Kind regards,

Magdalena Radulescu

Academic Editor

PLOS ONE

Journal Requirements:

Reviewers' comments:

Reviewer's Responses to Questions

**Comments to the Author**

1. If the authors have adequately addressed your comments raised in a previous round of review and you feel that this manuscript is now acceptable for publication, you may indicate that here to bypass the “Comments to the Author” section, enter your conflict of interest statement in the “Confidential to Editor” section, and submit your "Accept" recommendation.

Reviewer #2: All comments have been addressed

Reviewer #3: All comments have been addressed

2. Is the manuscript technically sound, and do the data support the conclusions?

Reviewer #2: Yes

Reviewer #3: Yes

3. Has the statistical analysis been performed appropriately and rigorously? 

Reviewer #2: Yes

Reviewer #3: Yes

4. Have the authors made all data underlying the findings in their manuscript fully available?

Reviewer #2: Yes

Reviewer #3: Yes

5. Is the manuscript presented in an intelligible fashion and written in standard English?

Reviewer #2: Yes

Reviewer #3: Yes

6. Review Comments to the Author

Reviewer #2: Thank you for allowing me to review your work. I believe this manuscript has potential to be published in PLOS ONE journal, but it requires significant improvements before submission. While the manuscript exhibits promise, it is crucial to address specific points to enhance the conceptual framework and logical coherence. Effectively addressing these areas will undoubtedly strengthen the overall quality of the document. ( "Renewable Energy as a Solution to Climate Change: Insights from a Comprehensive Study Across Nations" approaches.)

Comments

1.After reviewing this abstract, consider enhancing this abstract by incorporating sections on methods, results, and a dedicated conclusion for improved effectiveness.

2.Although the introduction is well-crafted, consider refining it by using the abbreviation (RE) for 'Renewable Energy.' Additionally, emphasize the concept of renewable energy projects and their potential to significantly reduce carbon emissions.

3.To improve the paper, this includes understanding what has been done, as well as what has not been done. Additionally, research questions and objectives should align with the purpose of the study.

4.While the authors adeptly address policy implications, it is recommended that they dedicate sufficient attention to articulating managerial implications. This addition would significantly contribute to enhancing the overall impact and effectiveness of their work. Additionally, you can take a look how this article has written managerial implication and limitation https://doi.org/10.3390/su151411289

Reviewer #3: In the latest edition, the authors effectively addressed all necessary comments, resulting in a further enhancement of the paper's quality.

7. PLOS authors have the option to publish the peer review history of their article (what does this mean?). If published, this will include your full peer review and any attached files.

Reviewer #2: **Yes: **Shabir Hussain Malik

Reviewer #3: No

---

## [Author Response · Author response to Decision Letter 2]

20 Jan 2024

Authors Response to editor and Reviewers Comments 

Dear Editor and Reviewers, 

We greatly appreciate the time invested in reading our manuscript and providing the necessary feedback to improve the overall quality of our work. All the mentioned comments have been considered and the revisions made accordingly. 

Please note that the line numbers referred to in this document are aligned with the revised manuscript which includes track changes.

Thank you once again for your valuable feedback.

Reviewer 2 comment 1: Thank you for allowing me to review your work. I believe this manuscript has potential to be published in PLOS ONE journal, but it requires significant improvements before submission. While the manuscript exhibits promise, it is crucial to address specific points to enhance the conceptual framework and logical coherence. Effectively addressing these areas will undoubtedly strengthen the overall quality of the document. ( "Renewable Energy as a Solution to Climate Change: Insights from a Comprehensive Study Across Nations" approaches.)

Comments

Authors’ Response to Reviewer 2 comment 1: Thank you for taking the time to review our work and provide your feedback on how our manuscript could be improved. We have carefully read and incorporated all the necessary changes to our paper improving the overall quality of the document.

Reviewer 2 comment 2: After reviewing this abstract, consider enhancing this abstract by incorporating sections on methods, results, and a dedicated conclusion for improved effectiveness.

Authors’ Response to Reviewer 2 comment 2: Well noted. According to the author guidelines of PLOS ONE, the abstract has not been divided into sub-sections. Instead, the abstract is written as a whole and does include the methods, results and conclusion in lines 28 -30 as follows: 

“We examined the extent of the impact of renewable energy on carbon dioxide over time using panel, linear, and non-linear regression approaches. The results of our analysis, revealed that the majority of countries with the exception of Canada, exhibited a downward trend, underscoring the potential of increasing renewable energy consumption as an effective method to reduce carbon dioxide emissions and combat climate change. Furthermore, to reduce emissions and combat climate change, it is advisable for nations with the highest carbon dioxide emissions to adopt and successfully transition to renewable energy sources.”

Reviewer 2 comment 3: Although the introduction is well-crafted, consider refining it by using the abbreviation (RE) for 'Renewable Energy.' Additionally, emphasize the concept of renewable energy projects and their potential to significantly reduce carbon emissions.

Authors’ Response to Reviewer 2 comment 3: Thank you for your feedback. The term Renewable Energy has been abbreviated using RE. The use of renewable energy projects to reduce carbon emissions has been added in the lines 89 - 94 as follows: 

“Governments and other relevant stakeholders initiate RE related projects to smoothen the transition from non-RE to RE in certain industries. It has been proven that stakeholder engagement and their relationship with project management teams contributes significantly to the success of the project in countries such as Pakistan [29]. A high success rate for such projects will directly impact the CO2 emission levels around those industries.”

Reviewer 2 comment 4: To improve the paper, this includes understanding what has been done, as well as what has not been done. Additionally, research questions and objectives should align with the purpose of the study.

Authors’ Response to Reviewer 2 comment 4: Duly noted. While the literature review presents a detailed overview on past studies, we have also included the following sentences in lines 112 – 119 in the introduction to clarify the research gap: 

“While previous research has focused on individual countries or a limited number of countries, there need to be more comprehensive global studies to provide a thorough understanding of the worldwide relationship between CO2 emissions and RE for a significant period. Therefore, this study provides a broader analysis based on a more substantial number of countries considering a more comprehensive period using the latest available data to obtain a more accurate depiction of the global status of the relationship between two variables helping for more precise decision-making.”

The research question and objective of the study have been mentioned in lines 115 – 119 as:

“Therefore, this study, satisfying the research question to understand what the impact of RE on CO2 emissions is, provides a broader analysis based on a more substantial number of countries considering a more comprehensive period using the latest available data to obtain a more accurate depiction of the global status of the relationship between two variables helping for more precise decision-making.”

“The study’s primary objective is to investigate the impact of RE on CO2 emissions in various countries …”

Reviewer 2 comment 5: While the authors adeptly address policy implications, it is recommended that they dedicate sufficient attention to articulating managerial implications. This addition would significantly contribute to enhancing the overall impact and effectiveness of their work. Additionally, you can take a look how this article has written managerial implication and limitation https://doi.org/10.3390/su151411289

Authors’ Response to Reviewer 2 comment 5: Highly appreciate the feedback. Taking your comment and the structure of the paper suggested into consideration we have added an additional section on managerial implications in the lines 557-568 as mentioned below.

“The top 12 CO2 emitting countries can gain many insightful recommendations from this study. Apart from Canada, where there’s a positive relationship from REC to CO2, all 11 other countries can highly promote RE related products in their most revenue generating industries. Countries such as India and China can focus on implementing RE projects to their manufacturing and production industry. Initiatives such as these projects will lead to sustainable revenue generating industries while generating opportunities for the community in the form of project managers, project management teams etc. Project managers will be further inspired to develop strategic plans to initiate high value collaborations [29]. With the involvement of the government, investors and other external stakeholders, renewable energy projects can bring an immense amount of value to these countries in their journey to achieving sustainable development.”

Reviewer 3 comment 3: In the latest edition, the authors effectively addressed all necessary comments, resulting in a further enhancement of the paper's quality.

Authors’ Response to Reviewer 3 comment 1: Thank you for your feedback and for taking the time to review our paper and provide precise recommendations to improve the overall quality of our work.

---

## [Decision Letter · Decision Letter 3]

16 Feb 2024

Renewable Energy as a Solution to Climate Change: Insights from a Comprehensive Study Across Nations

PONE-D-23-27314R3

Dear Dr. Jayathilaka,

We’re pleased to inform you that your manuscript has been judged scientifically suitable for publication and will be formally accepted for publication once it meets all outstanding technical requirements.

Kind regards,

Magdalena Radulescu

Academic Editor

PLOS ONE

Additional Editor Comments (optional):

Reviewers' comments:

Reviewer's Responses to Questions

**Comments to the Author**

1. If the authors have adequately addressed your comments raised in a previous round of review and you feel that this manuscript is now acceptable for publication, you may indicate that here to bypass the “Comments to the Author” section, enter your conflict of interest statement in the “Confidential to Editor” section, and submit your "Accept" recommendation.

Reviewer #2: All comments have been addressed

2. Is the manuscript technically sound, and do the data support the conclusions?

Reviewer #2: Yes

3. Has the statistical analysis been performed appropriately and rigorously? 

Reviewer #2: Yes

4. Have the authors made all data underlying the findings in their manuscript fully available?

Reviewer #2: Yes

5. Is the manuscript presented in an intelligible fashion and written in standard English?

Reviewer #2: Yes

6. Review Comments to the Author

Reviewer #2: Thank you for providing me with the opportunity to review this manuscript. The authors have effectively addressed all pertinent comments, leading to a notable improvement in the paper's quality. I am confident that this paper possesses the strength to be published in the PLOS ONE journal.

7. PLOS authors have the option to publish the peer review history of their article (what does this mean?). If published, this will include your full peer review and any attached files.

Reviewer #2: **Yes: **Shabir Hussain Malik

---

## [Editor Report · Acceptance letter]

23 May 2024

PONE-D-23-27314R3 

PLOS ONE

Dear Dr. Jayathilaka, 

I'm pleased to inform you that your manuscript has been deemed suitable for publication in PLOS ONE. Congratulations! Your manuscript is now being handed over to our production team.

Kind regards, 

on behalf of

Dr. Magdalena Radulescu 

%CORR_ED_EDITOR_ROLE%

PLOS ONE